# The microbiology of periprosthetic joint infections as revealed by sonicate cultures in Korea: Routine use of fungal and mycobacterial cultures is necessary?

Yoonjung Lee[1], Ahrang Lee[1], Hae Seong Jeong[1], Sung Un Shin[1], Uh Jin Kim[1,2☯*], Seong Eun Kim📷[1,2], Seung-Ji Kang[1,2], Sook-In Jung[1,2], Kyung-Soon Park[3], Jong Keun Seon[3], Jong-Hee Shin📷[4], Kyung-Hwa Park📷[1,2☯*]

1 Department of Infectious Diseases, Chonnam National University Hospital, Gwang-ju, Republic of Korea,
2 Department of Infectious Diseases, Chonnam National University Hospital Medical School, Gwang-ju, Republic of Korea, 3 Department of Orthopedic Surgery, Chonnam National University Hospital Medical School, Gwang-ju, Republic of Korea, 4 Department of Laboratory Medicine, Chonnam National University Hospital Medical School, Gwang-ju, Republic of Korea

☯ These authors contributed equally to this work.
* iammedkid@naver.com (K-HP); astralio@naver.com (UJK)

**Data Availability Statement:** The data are available from the Harvard Dataverse database (https://doi.org/10.7910/DVN/QNL0WW).

## Abstract

### Background

Although sonication is a valuable diagnostic tool for periprosthetic joint infections (PJI), it is not commonly utilized. We analyzed sonicate and intraoperative tissue culture results obtained from three hospitals to define the microbial etiology of PJIs in Korea. Furthermore, we investigated necessity of conducting regular fungal and mycobacterial cultures.

### Methods

We retrospectively analyzed data for patients with suspected orthopedic-related infections between 2017 and 2022, who had undergone prostheses removal surgery. We included 193 patients with suspected PJIs, and bacterial (n = 193), fungal (n = 193), and mycobacterial (n = 186) cultures were conducted on both sonicate and intraoperative tissue samples. The diagnosis of PJI was based on the European Bone and Joint Infection Society (EBJIS) criteria.

### Results

Out of 193 patients, 121 (62.7%) had positive sonicate cultures, while 112 (58.0%) had positive periprosthetic tissue cultures. According to EBJIS criteria, a total of 181 patients were diagnosed with PJI, and 141 patients received microbiological confirmation through sonicate fluid culture or tissue culture. Of the 181 patients, 28 were classified with acute PJI (within 3 months of implantation) and 153 with chronic PJI. Among 141 patients, *staphylococci* were the most common organisms, accounting for 51.8% of cases, followed by Gram-negative organisms (15.6%), fungus (8.5%), and mycobacteria (3.5%). Nearly 91.7% of fungal

**Funding:** This work was supported by the Chonnam National University Hospital Biomedical Research Institute (BCR124055) and GIST-CNUH research collaboration grant (BCR122060). The funding sources had no role in the study design, data collection and analysis, decision to publish, or preparation of the manuscript.

**Competing interests:** The authors have declared that no competing interests exist.

isolates were *Candida* species, which also grew in bacterial cultures. In total, 11 cases cultured positive only in tissue culture, whereas 20 cases cultured positive only in sonicate culture. The antibiotic treatment plans were adjusted according to culture results.

## Conclusions

Utilizing sonicate culture has greatly assisted in identifying pathogens responsible for chronic indolent PJIs, allowing suitable antimicrobial treatment. Based on few cases involving non-*Candida* and mycobacterial infections, it appears that routine fungal and mycobacterial cultures may not be necessary.

## Introduction

With the growing frequency of orthopedic surgeries, the occurrence of postoperative complications has increased, with infection being the most dreaded among them [1,2]. The rise in periprosthetic joint infection (PJI) caused by hard-to-treat microorganisms like antibiotic-resistant bacteria, fungi, and mycobacteria is imposing major economic burdens [3,4]. Therefore, it is crucial to have a highly reliable diagnosis of PJI and its microbiological epidemiology due to its invasive nature, the long duration of treatment it requires, and the limited options for antimicrobial therapy when dealing with challenging organisms. Extensive research has been conducted on the diagnostic criteria for PJI over the years, and these criteria encompass a combination of clinical, histological, and microbiological data. Notably, the clinical microbiology criteria outlined by various authoritative bodies such as the International Consensus Meeting (ICM) in 2018, the Infectious Disease Society of America (IDSA) in 2013, the American Academy of Orthopedic Surgeons (AAOS) in 2019, and the European Society for Bone and Joint Infections (EBJIS) in 2021 exhibit overlapping aspects, albeit with subtle distinctions [5–9]. Until now, the gold standard for microbiological diagnosis of PJI has been the culture of periprosthetic tissue, but the culture yields from these samples are low, posing a significant challenge in diagnosing PJI [10,11]. Sonication, on the other hand, effectively dislodges biofilms and the bacteria contained within them from the surfaces of implants; [12] hence, the EBJIS includes the culture of explanted prosthesis in their criteria.

The diagnosis of PJI traditionally involved obtaining multiple deep tissue samples during surgery. However, a recent practice has emerged where sonication cultures are used on removed devices [13]. The use of sonicate cultures in clinical microbiology laboratories of tertiary general hospitals is not yet widespread. Orthopedic surgeons often request multiple sets of fungal, mycobacterial, and traditional bacterial cultures. Since the microbial yields from sonicate and tissue samples vary significantly across different medical centers and countries, further research is necessary to investigate the microbiology of PJI and the diagnostic accuracy of sonicate cultures. To address this, we conducted a study examining the results of both sonicate and tissue cultures from three hospitals in Korea. Additionally, we evaluated the diagnostic utility of routine fungal and mycobacterial cultures of sonicate and tissue samples in identifying the cause of PJIs.

## Methods

### Study design and setting

This retrospective observational study was conducted at Chonnam National University Hospital, Hwasun Chonnam National University Hospital, and Bitgoeul Chonnam National

University Hospital. We made the registry of sonicate culture since October 2016. Patients undergoing prostheses removal surgery, whose prostheses were sent for sonication between January 2017 and December 2022, were analyzed. The medical records were abstracted by two of the authors (orthopedic surgeon and infectious disease specialist). Orthopedic prostheses were removed during the diagnosis and treatment steps of one- or two-stage surgery, as well as many other types of surgery. The exclusion criteria were fewer than two tissue samples sent for culture; prostheses not placed in appropriate, sterile plastic containers during transport; and implants subject to contamination during removal, transportation, or laboratory processing.

Patient demographics and comorbidities, operation sites, all previous orthopedic surgical procedures, clinical signs and symptoms, numbers of tissue specimens collected per patient, any use of antibiotics in the 28 days prior to prosthesis removal, and microbiological culture identifications were recorded. The numbers of conventional bacterial, fungal, and mycobacterial cultures performed for each case were examined, and the sensitivity and specificity of the culture rates, as well as the diagnostic performance of each culture type were calculated.

## Diagnosis of PJI

We suspected a PJI using modified clinical criteria (excluding microbiological results) used in previous studies [12,14] (S1 Table). Acute PJI refers to the onset of PJ infectious symptoms or signs within 3 months of implantation or surgery. Chronic PJI was characterized by persistent infectious symptoms or signs that typically presented > 3 months postoperatively.

Microbiological diagnoses proceeded as follows. All strains isolated from periprosthetic tissues/pus, synovial fluid, and prosthesis sonicate cultures were recorded. According to the IDSA guidelines [7], when the diagnostic standards for orthopedic infections are met, a virulent microorganism (e.g., *Staphylococcus aureus*) isolated from even a single specimen is considered to be the causative organism. For low-virulence pathogens and/or potential contaminants such as coagulase-negative staphylococci (CoNS), *Corynebacterium* species, or *Cutibacterium acnes*, at least two culture-positive perioperative and preoperative samples were required for diagnosis. We considered a case to be a true fungal or mycobacterial infection if the treating clinician prescribed antifungal or antimycobacterial agents following isolation of a fungal or mycobacterial organism. The sensitivity profiles of all strains were determined as described by the Clinical Laboratory Standards Institute [15]. The diagnostic criteria of the EBJIS, including the microbiological results (S1 Table), were applied for all other assessments [5].

## Microbiological procedures

Microbiological studies were performed from preoperative synovial fluid or intraoperative periprosthetic tissue and sonicate fluid. Bacterial culture, fungal culture or mycobacterial culture of each specimen were ordered. In the laboratory, samples for bacterial culture were inoculated onto blood agar plates (BAPs), chocolate agar plates, and MacConkey agar plates in jars at 35°C, and also into anaerobic thioglycolate broth. The agar plates were incubated at 35–37°C for 5 days aerobically and 14 days anaerobically. Thioglycollate broths were incubated for 14 days at 35–37°C; in the event of bacterial growth (turbidity), the liquid was seeded onto BAPs (both aerobic and anaerobic cultures). Saboured dextrose agar and potato dextrose agar plates were used for fungal culture. For mycobacterial culture, pretreated samples were inoculated into mycobacterium growth indicator tubes (MGIT), incubated in the MGIT 960 device for 1 week, and then cultured for 6 weeks. The samples were also inoculated into 3% (w/v) Ogawa medium, incubated for 1 week, and then cultured for 6 weeks.

**1) Preoperative synovial fluid culture.**   0.1 mL of synovial fluid was inoculated onto plates and into thioglycolate broth, and aerobic and anaerobic growth assessed.

**2) Intraoperative periprosthetic tissue cultures.**   The surgeon selected several representative tissue samples from the surgical field; most were inflamed or purulent. A complete bacterial and fungal culture setup included five aerobic agar plates, one enrichment broth, three anaerobic agar plates, a gram stain, and a KOH stain. At least two complete culture setups were ordered for each patient. The evaluation of up to eight additional bacteria and fungi was possible at the physician's discretion. Mycobacterial culture and MTB-PCR hybridization were added if the surgeon so requested.

**3) Sonicate cultures.**   Prostheses explanted during surgical procedures were processed using the Mayo Clinic protocol [16,17]. The prostheses were placed in sterile containers with 400 mL of Ringer's solution. After vortexing for 30 s, the containers were sonicated at 40 kHz for 5 min, followed by vortexing for a further 30 s. The sonication fluid was transferred to a 50-mL tube and centrifuged for 5 min. The supernatant was removed and aliquots of 0.1 mL inoculated onto aerobic and anaerobic BAPs (0.1 mL inoculum equals 10 mL of the original sonicated sample). Sonicate cultures were cultured at 35–37°C for 5 days aerobically and 14 days anaerobically, and the numbers and identities of all colonies recorded. If a specimen contained $\geq$ 5 colony-forming units (CFU) of any organism, identification and susceptibility tests were performed. For sonicate fluids, we considered that a culture was positive if growth exceeded 20 CFU/10 mL according to mayo clinic protocol [18], with the exception of virulent microorganisms such as *S. aureus*, for which any growth was considered positive. The complete bacterial culture setup included two BAPs per sonication fluid. Additional culture work-ups for fungi or mycobacteria were at the surgeon's discretion (no guidelines).

## Statistical analyses

The baseline characteristics of all groups were compared using the chi-squared test. All calculations were performed using the SPSS v28 statistical software package (SPSS Inc., Chicago, IL, USA) and R v4.3.2 (R Core Team, Vienna, Austria).

## Ethics statement

The present study protocol was reviewed and approved by the Institutional Review Board of Chonnam National University Hospital (approval No. 2023–091) and the need for informed consent was waived.

## Results

### Study population

During the study period, 333 patients underwent orthopedic implant removal for various reasons including suspected PJI, aseptic loosening, or fixation failure. Revision surgeries after treatment to rule out persistent infections were excluded (Fig 1). Of the 333 patients, 193 undergoing revisions because of suspected prosthetic hip or knee joint infections and whose removed prostheses underwent appropriate sonicate and tissue cultures were included. Fungal tissue cultures were performed for all 193 patients and mycobacterial cultures for 186 patients.

Table 1 lists the patient characteristics. Overall, the median age was 71.2 years and 46.6% were male. Patients with knee prostheses comprised 55.4% of all cases and preoperative antibiotics were administered to 46.1% of the 193 cases. A total of 162 patients (83.9%) had chronic PJIs. Prosthesis removal during a two-stage exchange operation was the most common form of retrieval. Preoperative synovial fluids were cultured for 137 patients; 64 were positive.

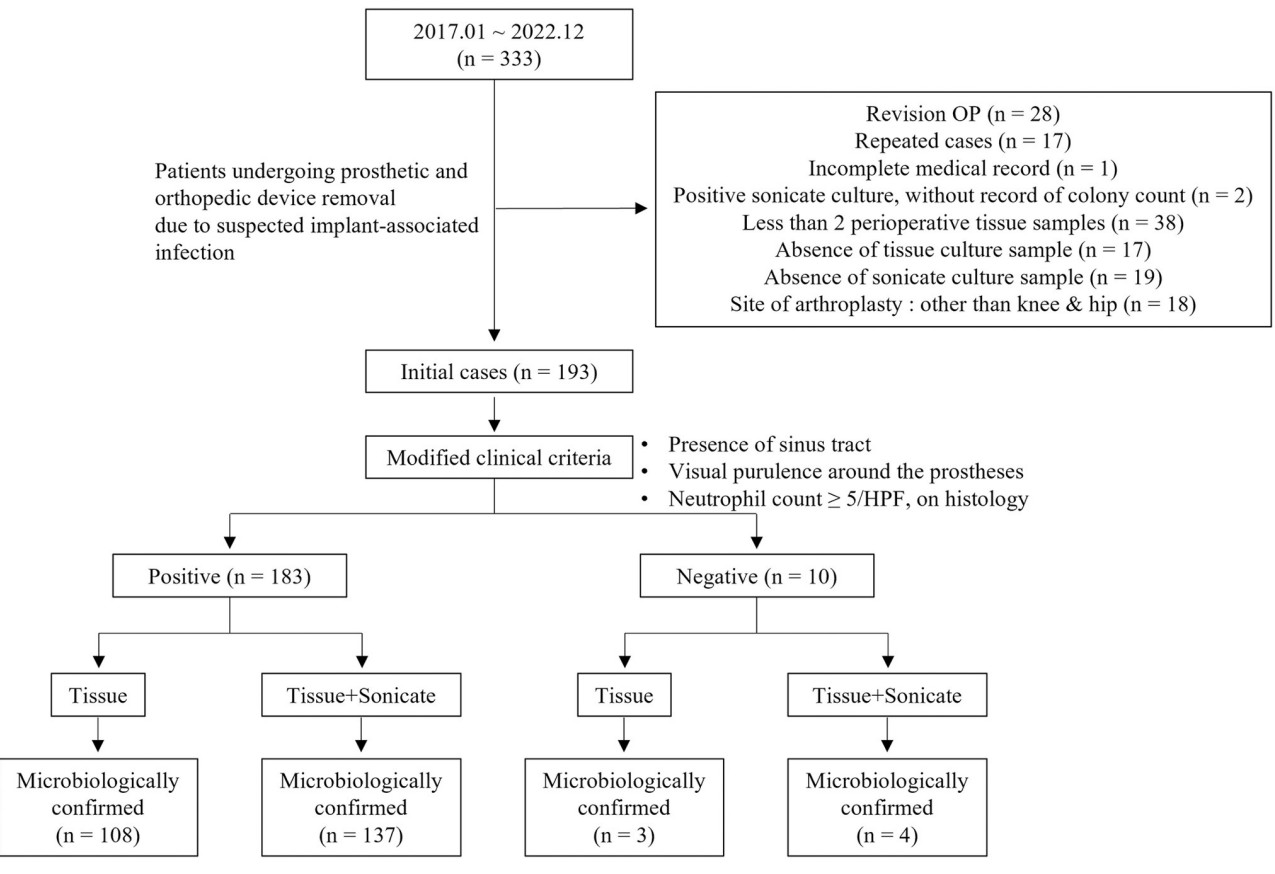

**Fig 1. Flow chart.**

Sonicate cultures were positive in 121 of the 193 patients (62.7%) and intraoperative tissue cultures were positive in 112 (58.0%). Each patient yielded between two and eight tissue culture samples (median 3.69 samples) and 28.5% more than five. When the modified clinical criteria were applied, 183 were diagnosed with PJI. Of all patients assessed, 93.8% (181/193) were diagnosed with PJIs using the EBJIS criteria. Fig 1 show inclusion criteria and number of microbiologically confirmed cases including preoperative synovial fluid culture.

## Microbiological assessment

Table 2 lists the microbiological results for 181 patients with PJI diagnoses using the 2021 EBJIS criteria. A total of 141 patients was microbiologically confirmed. A total of 112 patients (61.9%) yielded positive tissue cultures and 121 (66.9%) positive sonicate cultures. In most cases, only one strain was identified as a causative organism. The most commonly cultivated organisms were staphylococci (51.8%), followed by Gram-negative organisms (15.6%). In terms of CoNS, the sonicate cultures yielded more positive results than did tissue cultures (27.6% vs. 21.0%), but *S. aureus* was more frequently detected in tissue culture. Of Gram-negative organisms, *Escherichia coli* was the most common; sonicate cultures were more often positive than others. *Pseudomonas aeruginosa* was detected at a higher rate in tissue culture. Twelve *Candida* species and one mold grew in culture. When acute and chronic PJI were

**Table 1. Demographic and clinical characteristics of 193 patients with suspected prosthetic joint infections.**

| Characteristic | Value | n (%) |
|---|---|---|
| Mean age, years (range) | 71.2 (42.0–89.0) | 193 |
| Sex | Male | 90 (46.6) |
| Sinus tract status | Present | 27 (14) |
| Visual purulence | Present | 175 (90.7) |
| Permanent biopsy, n = 76 | Neutrophil≥5/HPF | 56 (73.6) |
| Frozen biopsy, n = 50 | Neutrophil≥5/HPF | 32 (64) |
| Site of arthroplasty | Hip | 86 (44.6) |
| | Knee | 107 (55.4) |
| Preoperative antimicrobial use within 28 days | Yes | 89 (46.1) |
| Acute/chronic PJI | Acute | 31 (16.1) |
| | Chronic | 162 (83.9) |
| Operation type | Debridement and implant retention | 20 (10.4) |
| | One-stage exchange | 13 (6.7) |
| | Two-stage exchange | 153 (79.3) |
| | Girdlestone operation | 6 (3.1) |
| | Arthrodesis | 1 (0.5) |
| Serum C-reactive protein, mg/dL | > 10 | 42 (21.8) |
| | 1–10 | 125 (64.8) |
| | ≤ 1 | 26 (13.5) |
| Erythrocyte sedimentation rate, mm/h, n = 182 | > 30 | 159 (87.4) |
| | ≤ 30 | 23 (12.6) |
| Preoperative synovial fluid culture, n = 137 | Positive | 64 (46.7) |
| Intraoperative tissue culture, n = 193 | Positive | 112 (58.0) |
| Intraoperative sonicate fluid culture, n = 193 | Positive | 121 (62.7) |
| Number of periprosthetic tissue samples taken | ≥ 5 | 55 (28.5) |
| | 2–4 | 138 (71.5) |
| 2021 EBJIS Criteria | Confirmed and likely | 181 (93.8) |
| | Unlikely | 12 (6.2) |

n, number; EBJIS, European Bone and Joint Infection Society.

analyzed separately, CoNS and *E. coli* were detected more frequently in sonicate cultures from patients with chronic PJIs.

Table 3 lists cases for which the results differed between periprosthetic tissue and sonicate cultures. Twenty PJIs were detected by sonication fluid cultures but not tissue cultures, and the antibiotic regimens were changed to reflect the sonicate culture results. Among 20 cases, 16 cases were patients with an isolated positive sonicate culture. The sensitivity increased when sonicate culture were combined with intraoperative tissue or preoperative synovial fluid cultures. Eleven cases were only tissue-culture positive. Two patients were diagnosed with polymicrobial infections via tissue culture.

## Fungal and mycobacterial cultures

A total of 713 conventional bacterial cultures, 713 fungal stain/cultures, and 331 mycobacterial cultures from the intraoperative tissues of 193 patients with suspected PJIs were performed. The test positivity rates of fungal and mycobacterial cultures were 2.95% (21/713) and 3.02%

**Table 2. Microbiological etiologies of 181 patients with periprosthetic joint infections diagnosed using the 2021 EBJIS criteria.**

| | Synovial fluid (n = 132) | | Total (n = 181) | | | | Acute/Chronic PJI | | | | | | | |
| | | | | | | | Acute PJI (n = 28) | | | | Chronic PJI (n = 153) | | | |
| Microbiology | n | % | Tissue | | Sonicate | | Tissue | | Sonicate | | Tissue | | Sonicate | |
| | | | n | % | n | % | n | % | n | % | n | % | n | % |
|---|---|---|---|---|---|---|---|---|---|---|---|---|---|---|
| Number of detected organisms (n = 141) | 64 | | 112 | | 121 | | | | | | | | | |
| Bacteria | | | | | | | | | | | | | | |
| Gram positive organisms (n = 95) | | | | | | | | | | | | | | |
| *Staphylococcus aureus* (n = 24) | 18 | 13.6% | 21 | 11.6% | 18 | 9.9% | 3 | 10.7% | 2 | 7.1% | 18 | 11.8% | 16 | 10.5% |
| Coagulase negative *staphylococci* | | | | | | | | | | | | | | |
| *S. epidermidis* (n = 37) | 12 | 9.1% | 29 | 16.0% | 37 | 20.4% | 3 | 10.7% | 3 | 10.7% | 26 | 17.0% | 34 | 22.2% |
| Other coagulase negative *staphylococci* (n = 12) | 5 | 3.8% | 9 | 5.0% | 13 | 7.2% | 1 | 3.6% | 2 | 7.1% | 8 | 5.3% | 11 | 7.2% |
| *Corynebacterium striatum* (n = 4) | 1 | 0.8% | 2 | 1.1% | 4 | 2.2% | - | - | 1 | 3.6% | 2 | 1.3% | 3 | 2.0% |
| Enterococci (n = 9) | | | | | | | | | | | | | | |
| *E. faecalis* (n = 5) | 2 | 1.5% | 4 | 2.2% | 4 | 2.2% | - | - | - | - | 4 | 2.6% | 4 | 2.6% |
| *E. faecium* (n = 4) | - | - | 4 | 2.2% | 4 | 2.2% | 1 | 3.6% | 1 | 3.6% | 3 | 2.0% | 3 | 2.0% |
| Streptococci (n = 8) | | | | | | | | | | | | | | |
| *S. aglactiae* (n = 4) | 3 | 2.3% | 2 | 1.1% | 3 | 1.7% | - | - | - | - | 2 | 1.3% | 3 | 2.0% |
| *S. gordonii* (n = 1) | 1 | 0.8% | 1 | 0.6% | 1 | 0.6% | - | - | - | - | 1 | 0.7% | 1 | 0.7% |
| *S. mitis/oralis* (n = 2) | - | - | 2 | 1.1% | 2 | 1.1% | - | - | - | - | 2 | 1.3% | 2 | 1.3% |
| *S. mutans* (n = 1) | - | - | 1 | 0.6% | 1 | 0.6% | - | - | - | - | 1 | 0.7% | 1 | 0.7% |
| Etc. | | | | | | | | | | | | | | |
| *Erysipelothrix rhusiopathiae* (n = 1) | 1 | 0.8% | 1 | 0.6% | 1 | 0.6% | - | - | - | - | 1 | 0.7% | 1 | 0.7% |
| Gram negative organisms (n = 22) | | | | | | | | | | | | | | |
| *Escherichia coli* (n = 12) | 8 | 6.1% | 10 | 5.5% | 13 | 7.2% | 2 | 7.1% | 3 | 10.7% | 8 | 5.2% | 10 | 6.5% |
| *Enterobacter cloacae* (n = 1) | - | - | 1 | 0.6% | 1 | 0.6% | - | - | - | - | 1 | 0.7% | 1 | 0.7% |
| *Enterobacter aerogenes* (n = 1) | - | - | - | - | 1 | 0.6% | - | - | - | - | - | - | 1 | 0.7% |
| *Klebsiella oxytoca* (n = 1) | 1 | 0.8% | - | - | - | - | - | - | - | - | - | - | - | - |
| *Klebsiella pneumoniae* (n = 1) | 1 | 0.8% | - | - | - | - | - | - | - | - | - | - | - | - |
| *Pseudomonas aeruginosa* (n = 4) | - | - | 4 | 2.2% | 2 | 1.1% | 1 | 3.6% | 0 | 0.0% | 3 | 2.0% | 2 | 1.3% |
| *Acinetobacter baumannii* (n = 1) | - | - | 1 | 0.6% | 1 | 0.6% | 1 | 3.6% | 1 | 3.6% | 1 | 0.7% | 1 | 0.7% |
| *Alcaligenes xylosoxidans* (n = 1) | - | - | 1 | 0.6% | 1 | 0.6% | - | - | - | - | - | - | - | - |
| Fungi | | | | | | | | | | | | | | |
| Yeast: *Candida* species (n = 11) | | | | | | | | | | | | | | |
| *C. albicans* (n = 5) | 1 | 0.8% | 4 | 2.2% | 3 | 1.7% | 2 | 7.1% | 1 | 3.6% | 2 | 1.3% | 2 | 1.3% |
| *C. parapsilosis* (n = 4) [a] | 2 | 1.5% | 4 | 2.2% | 3 | 1.7% | - | - | - | - | 4 | 2.6% | 3 | 2.0% |
| *C. pelliculosa* (n = 2) | 1 | 0.8% | 1 | 0.6% | 2 | 1.1% | - | - | - | - | 1 | 0.7% | 2 | 1.3% |
| Mold: *Lomentospora prolificans* (n = 1) | 1 | 0.8% | 1 | 0.6% | 1 | 0.6% | - | - | - | - | 1 | 0.7% | 1 | 0.7% |
| *Mycobacterium* species (n = 4) | | | | | | | | | | | | | | |
| *M. tuberculosis* (n = 2) | 2 | 1.5% | 2 | 1.1% | - | - | - | - | - | - | 2 | 1.3% | - | - |
| Nontuberculous mycobacteria | | | | | | | | | | | | | | |
| *M. fortuitum* (n = 1) | 1 | 0.8% | 0 | 0.0% | 1 | 0.6% | - | - | - | - | - | - | 1 | 0.7% |
| *M. terrae complex* (n = 1) | - | - | 1 | 0.6% | - | - | - | - | - | - | 1 | 0.7% | - | - |
| Polymicrobial infection (n = 8) | 3[b] | 2.3% | 6[c] | 3.3% | 4[d] | 2.2% | 2 | 7.1% | - | - | 4 | 2.6% | 4 | 2.6% |

(*Continued*)

**Table 2.** (Continued)

| Microbiology | Synovial fluid (n = 132) | | Total (n = 181) | | | | Acute/Chronic PJI | | | | | | | |
| | | | | | | | Acute PJI (n = 28) | | | | Chronic PJI (n = 153) | | | |
| | | | Tissue | | Sonicate | | Tissue | | Sonicate | | Tissue | | Sonicate | |
| | n | % | n | % | n | % | n | % | n | % | n | % | n | % |
|---|---|---|---|---|---|---|---|---|---|---|---|---|---|---|
| No pathogen detected (n = 40) | 68 | 51.5% | 69 | 38.1% | 60 | 33.1% | 12 | 42.9% | 14 | 50.0% | 57 | 37.3% | 46 | 30.1% |

n, number; EBJIS, European Bone and Joint Infection Society; PJI, Periprosthetic Joint Infection.

[a]One *C. parapsilosis* was cultured with *K. pneumoniae* in synovial fluid and one *C. parapsilosis* was cultured in synovial fluid, tissue, sonicate.

[b]*K. pneumoniae* + *E. cloacae*, *P. aeruginosa* + *A. baumannii*, *C.parapsilosis* + *E. faecalis* + *S. aureus*.

[c]*K. pneumoniae* + *E. cloacae*, *S. dysgalactiae* + *S. agalactiae*, *S. gordonii* + *S. mitis/oralis*, *E. faecalis* + *K. oxytoca*, *Staphylococcus lugdunensis* + *E. faecalis*, *C. striatum* + *M. fortuitum*.

[d]*K. pneumoniea* + *E. cloacae*, *S. dysgalactiae* + *S. agalactiae*, *S. gordonii* + *S. mitis/oralis*, *E. faecalis* + *K. oxytoca*.

(10/331) respectively. Thirteen cultures were confirmed to be true fungal infections. Surgeons wrote fungal culture orders for synovial fluids (n = 137 patients, 232 fungal cultures) and sonicate fluids (n = 193 patients, 113 fungal cultures). Of five patients with mycobacterial infections, mycobacteria grew in four mycobacterial cultures. Table 4 lists the true fungal PJI culture results. *Candida* species were predominant; all fungal pathogens grew in bacterial cultures and there were only three positive fungal staining results. In 3 out of 5 cases in which mycobacteria were identified and in 7 out of 13 cases in which fungus was identified, multiple revision operations were performed.

**Table 3. Cases in which periprosthetic tissue and sonicate culture results differed.**

| Case classifications | Periprosthetic tissue culture organisms | Sonicate culture organisms | No. cases | An isolated positive culture[a] |
|---|---|---|---|---|
| Positive sonicate cultures and negative periprosthetic tissue cultures (n = 20) | – | *Staphylococcus epidermidis* | 8 | 7 |
| | – | Coagulase-negative *staphylococci* | 3 | 2 |
| | – | *Corynebacterium striatum* | 1 | 1 |
| | – | *Streptococcus agalactiae* | 1 | 1 |
| | – | *Escherichia coli* | 3 | 2 |
| | – | *Enterobacter aerogenes* | 1 | 1 |
| | – | *Candida albicans* | 1 | 1 |
| | – | *Candida pelliculosa* | 1 | 1 |
| | – | *Mycobacterium fortuitum* | 1 | 0 |
| Negative sonicate cultures and positive periprosthetic tissue cultures (n = 11) | *Staphylococcus aureus* | – | 3 | 0 |
| | *Pseudomonas aeruginosa* | – | 2 | 2 |
| | *Candida albicans* | – | 2 | 1 |
| | *Candida parapsilosis* | – | 1 | 1 |
| | *Mycobacterium tuberculosis* | – | 2 | 0 |
| | *Mycobacterium terrae* complex | – | 1 | 1 |
| Discordant (positive) sonicate cultures and periprosthetic tissue cultures (n = 2) | *Staphylococcus lugdunensis* + *Enterococcus faecalis* | *S. lugdunensis* | 1 | - |
| | *C. striatum* + *M. fortuitum* | *C. striatum* | 1 | - |

[a]An isolated positive culture means positive culture results in exclusively sonicate or tissue culture.

**Table 4. Culture results for fungal periprosthetic joint infections.**

| Case no. | Causative organism | Synovial fluid cultures | | Periprosthetic tissue cultures | | Sonicate cultures | |
|---|---|---|---|---|---|---|---|
| | | Positive culture no./ no. of bacterial cultures | Positive culture no./ no. of fungal cultures | Positive culture no./ no. of bacterial cultures | Positive culture no./ no. of fungal cultures | Positive culture no./ no. of bacterial cultures | Positive culture no./ no. of fungal cultures |
| 1 | *Candida albicans* | 0/1 | 0/1 | 2/4 | 2/4 | 0/1 | 0/1 |
| 2 | *Candida albicans* | – | – | 2/2 | 2/2 | 1/1 | 1/1 |
| 3 | *Candida albicans* | – | – | 1/2 | 1/2 | 1/1 | - |
| 4 | *Candida albicans* | – | – | 0/3 | 0/3 | 1/1 | 0/1 |
| 5 | *Candida albicans* | 1/1 | 0/1 | 2/2 | 0/2 | 0/1 | 0/1 |
| 6 | *Candida parapsilosis* | 0/1 | 0/1 | 1/2 | 1/2 | 0/1 | - |
| 7 | *Candida parapsilosis* | 0/1 | 0/1 | 2/3 | 2/3 | 1/1 | - |
| 8 | *Candida parapsilosis* | 1/2 | 2/2 | 3/4 | 3/4 | 1/1 | 1/1 |
| 9 | *Candida pelliculosa* | 1/3 | 2/2 | 2/6 | 4/6 | 1/1 | - |
| 10 | *Candida pelliculosa* | 0/2 | 0/2 | 0/5 | 0/5 | 1/1 | - |
| 11 | *Lomentospora prolificans* | 1/3 | 1/3 | 1/4 | 2/4 | 1/1 | - |
| 12 | *Polymicrobial infection*[a] | 0[b]/2 | 2[c]/2 | 4[c]/4 | 4[c]/4 | 1/1 | 1/1 |
| 13 | *Polymicrobial infection*[d] | 2/3 | 2/3 | 0/4 | 0/4 | 0/1 | 0/1 |

no, number.

[a]Klebsiella pneumoniae, C. parapsilosis.

[b]*K. pneumoniae* was cultured in one of two paired bacterial cultures.

[c]C. parapsilosis.

[d]Enterococcus faecalis, Staphylococcus aureus, C. parapsilosis.

## Discussion

We analyzed 193 patients whose prostheses were removed because of suspected hip or knee infections. Of all patients assessed, 93.8% (181/193) were diagnosed with PJIs using the EBJIS criteria. Among microbiologically confirmed 141 PJI cases, the most common organisms were staphylococci (51.8%) and Gram-negative organisms constituted 15.6% of all pathogens. Sonicate fluid cultures detected 20 microorganisms that did not grow in periprosthetic tissue cultures. The 13 fungal pathogens all grew in bacterial cultures. Only five patients had mycobacterial infections.

CoNS were frequently detected in sonicate cultures; most PJI CoNS infections were chronic. The ability of bacteria to form biofilms on the surfaces of prostheses contributes greatly to chronic PJI and is one of the main causes why intraoperative tissue samples are often not positive [19]. Ultrasound-mediated dislodgement of biofilms from the surfaces of removed prostheses increases the sensitivities of microbiological studies that seek to identify underlying pathogens [20,21]. Here, *S. aureus* and *P. aeruginosa* were detected more frequently in tissue culture, perhaps because these organisms are both more virulent than others and invasive. However, intraoperative tissue cultures alone are compromised by high rates of contamination and thus false-positive results [22]. Generally, microbiology laboratories report only positive

or negative growth; quantitation is lacking. When normal skin flora such as CoNS and *Corynebacterium* species are identified in preoperative synovial fluid or periprosthetic tissue culture, sonicate culture aids pathogen identification by yielding quantitative information.

Although sonication is technically simple, most microbiology laboratories of Korean hospitals do not yet perform sonication cultures. We found that combined sonicate, intraoperative tissue, and synovial fluid cultures increased sensitivity. When a consensus is reached to the effect that isolation of phenotypically identical microorganisms from more than one culture is the gold standard for PJI diagnosis, sonicate culture should be included in PJI diagnosis in Korea.

Most PJIs are caused by bacteria [23,24]. The ICM guideline recommends both fungal and mycobacterial cultures for immunocompromised patients, those with previously confirmed infections, and arthroplasty patients with culture-negative joints [25]. Tai et al. [26] prepared indications for fungal culture when diagnosing PJIs. These include immunocompromised patients (i.e., those undergoing solid organ or hematopoietic stem cell transplantation, patients with AIDS, and cancer patients on active chemotherapy); patients with a history of fungal or mycobacterial PJI; a PJI in a setting of a disseminated fungal or mycobacterial infection; and recurrent culture-negative PJI despite the use of appropriate bacterial culture techniques and adequate treatment. We found that orthopedic surgeons routinely ordered fungal cultures and fungal staining of intraoperative tissues, synovial fluids, or sonicates. In the present study, patients with true fungal PJIs were immunocompetent, but *Candida* species were readily detected on bacterial culture and only three positive fungal staining results were noted. Such staining should be discouraged when diagnosing PJI; more research on the specific diagnostic evaluation of non-candidal fungal PJIs is needed.

Tuberculosis is common in Korea; [27] mycobacterial culture may be necessary depending on the clinical course of the patient. In this study, only five cases had mycobacterial infections, but surgeons ordered mycobacterial cultures for > 96% of suspected PJI patients. Routine performance of specialized cultures increases the financial burden on patients and incurs unnecessary medical expenses [28,29]. Any need for such cultures should be carefully considered on the basis of the risk factors, patient history, and clinical progress in the detection of culture-negative PJIs.

Our work had several limitations. First, the overall culture sensitivity was lower than those of other studies [12,13,30,31]. In our work, about 46% of patients received antibiotics treatment before operation and 70% of patients yielded ≤ 4 tissue samples. Some tissue samples were obtained after irrigation of wounds or deeper tissues. Such variations are inevitable in any retrospective study. Second, we included both hip and knee PJIs. Third, some serum biomarker and synovial fluid data required for PJI diagnosis were missing. Forth, we recorded only a few true mycobacterial PJI and mold infections. More clinical research on hard-to-treat microorganisms is required; specific diagnostic evaluations are needed. Lastly, there were 16 cases with an isolated positive sonicate culture and a significant number were low virulent microorganisms in our work. Although Rondaan C, *et al*. recently, reported the clinical relevance of an isolated positive sonicate culture [32], these findings require further exploration for clinical importance.

## Conclusions

In summary, a combination of periprosthetic tissue, sonicate, and synovial fluid cultures improved PJI pathogen identifications and will aid the choice of appropriate antibiotics. Sonicate culture is simple; general laboratories should add this valuable technique to the microbiological diagnosis of PJIs in Korea. *Candida* infections were well detected in bacterial cultures.

Fungal staining/culture and mycobacterial culture are of little use in terms of PJI diagnosis; fungal and mycobacterial cultures may be required when indicated.

## Supporting information

**S1 Table. Diagnostic criteria for hip or knee periprosthetic joint infection.**
(DOCX)

## Acknowledgments

We wish to thank participating care units for their collaboration in the realization of this study.

## Author Contributions

**Conceptualization:** Uh Jin Kim, Kyung-Hwa Park.

**Data curation:** Yoonjung Lee, Ahrang Lee, Hae Seong Jeong, Sung Un Shin, Uh Jin Kim, Seong Eun Kim, Seung-Ji Kang, Sook-In Jung, Kyung-Soon Park, Jong Keun Seon, Jong-Hee Shin.

**Formal analysis:** Yoonjung Lee, Uh Jin Kim, Kyung-Hwa Park.

**Investigation:** Yoonjung Lee, Uh Jin Kim.

**Methodology:** Yoonjung Lee, Uh Jin Kim, Jong-Hee Shin, Kyung-Hwa Park.

**Visualization:** Yoonjung Lee.

**Writing – original draft:** Yoonjung Lee, Kyung-Hwa Park.

**Writing – review & editing:** Yoonjung Lee, Uh Jin Kim, Jong-Hee Shin, Kyung-Hwa Park.

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
