## [Decision Letter · Decision Letter 0]

24 Mar 2024

PONE-D-24-06916The microbiology of prosthetic joint infections as revealed by sonicate cultures in Korea: Routine use of fungal and mycobacterial cultures is necessary?PLOS ONE

Dear Dr. Park,

Thank you for submitting your manuscript to PLOS ONE. After careful consideration, we feel that it has merit but does not fully meet PLOS ONE’s publication criteria as it currently stands. Therefore, we invite you to submit a revised version of the manuscript that addresses the points raised during the review process.

We look forward to receiving your revised manuscript.

Kind regards,

Abbas Farmany

Academic Editor

PLOS ONE

Journal Requirements:

"This work was supported by the the Chonnam National University Hospital Biomedical Research Institute (BCR122060) GIST-CNUH research collaboration grant funded by the CNUH in 2023. "

Reviewers' comments:

Reviewer's Responses to Questions

**Comments to the Author**

1. Is the manuscript technically sound, and do the data support the conclusions?

Reviewer #1: Partly

Reviewer #2: Yes

2. Has the statistical analysis been performed appropriately and rigorously? 

Reviewer #1: N/A

Reviewer #2: Yes

3. Have the authors made all data underlying the findings in their manuscript fully available?

Reviewer #1: No

Reviewer #2: Yes

4. Is the manuscript presented in an intelligible fashion and written in standard English?

Reviewer #1: Yes

Reviewer #2: No

5. Review Comments to the Author

Reviewer #1: This is an important topic especially since we need a diverse voice in Ortho research. However, it needs further work before being ready for publication.

Major comments:

1. It’s confusing what are the primary objectives of this study. It seems like it is focusing on the performance and results of sonicate cultures, fungal and mycobacterial cultures. However, it also talks about the overall microbiology of PJI in Table 2, then sensitivity/specificity of microbiologic testing in Table 4. I recommend focusing on the paper on sonicate culture, fungal and mycobacterial culture results. For the gold standard, the authors could focus on EBJIS criteria instead of adding the ICM criteria. The accuracy of synovial and periprosthetic tissue cultures is another broad topic which can be addressed using another strategy and not appropriate to be linked to this paper.

Minor comments:

1. Recommend editing Introduction and abstract to be more general in nature for broader readership rather than focusing on Korea. Suggest, “The use of sonicate cultures in clinical microbiology laboratories of tertiary general hospitals is not yet widespread. Orthopedic surgeons often request multiple sets of fungal, mycobacterial, and traditional bacterial cultures. Since the microbial yields from sonicate and tissue samples vary significantly across different medical centers and countries, further research is necessary to investigate the microbiology of PJI and the effectiveness of sonicate cultures”

2. Edit terms “staphylococci” – Small letter s, not capital. “Gram-negative”- Capital letter G.

3. Harmonize "periprosthetic joint infections" instead of "prosthetic"

4. Sonication was not part of the IDSA diagnostic criteria, please revise.

5. Recommend changing term, “efficacy” to “accuracy” and “utility” in introduction. Diagnostic tests are not judged based on efficacy. It’s a term for therapeutic tests.

Reviewer #2: The manuscript titled "The microbiology of prosthetic joint infections as revealed by sonicate cultures in Korea: Routine use of fungal and mycobacterial cultures is necessary?" focuses on interesting topic. I commend the authors for their dedication and express my appreciation for the chance to evaluate their manuscript. The manuscript is crafted in a decent manner, that can in terms on English be also improved. Congratulations to the authors on their findings. A retrospective multi-centre study that in my opining should be taken into consideration to be published after a revision. The conclusions are consistent with the evidence and arguments presented and effectively address the main question posed.

After assessing the manuscript, the following issues raised my concerns or represent suggestions that from my point of view can increases the overall quality of the manuscript:

- Abstract – results – “According to EBJIS criteria, total of 181 patients were diagnosed with acute (28 patients) or chronic (153 patients) PJIs” – The EBJIS criteria represent a set of definition criteria for periprosthetic joint infections (PJIs), whereas in 2004 for e.g., Zimmer et al proposed a classification system for PJIs, which was subsequently published in the New England Journal of Medicine. It is important to note that while the EBJIS criteria serve to define PJIs, the classification system proposed by Zimmer et al categorizes PJIs into distinct groups based on their clinical presentation and various other factors. Please also note the statements of the authors of the EBJIS criteria “This definition does not distinguish PJI on the basis of the duration of the infection (acute or chronic) or the time of onset from implantation (early or late). These terms are not defined with any degree of certainty with time-dependant cut-offs, and so cannot be included in a definition of PJI.”

- Abstract – “In total, 11 cases cultured positive only in tissue culture, whereas 20 cases cultured positive only in sonicate culture.” What is your approach in cases of only positive sonication fluid cultures? As a suggestion – take a look at this multicenter retrospective cohort study - Rondaan C, Maso A, Birlutiu RM, Fernandez Sampedro M, Soriano A, Diaz de Brito V, Gómez Junyent J, Del Toro MD, Hofstaetter JG, Salles MJ, Esteban J, Wouthuyzen-Bakker M; ESCMID Study Group on Implant Associated Infections (ESGIAI). Is an isolated positive sonication fluid culture in revision arthroplasties clinically relevant? Clin Microbiol Infect. 2023 Jul 28:S1198-743X(23)00345-2. doi: 10.1016/j.cmi.2023.07.018. Epub ahead of print. PMID: 37516385.

- Diagnosis of PJIs- “To evaluate the diagnostic utility of periprosthetic tissue and sonicate cultures, PJIs were diagnosed using modifications of clinical criteria (excluding microbiological results) used in previous studies [12, 16].” Do you have some internal validation study for this criteria?

- Sonication cultures – “For sonicate fluids, we considered that a culture was positive if growth exceeded 20 CFU/10 mL, with the exception of virulent microorganisms such as S. aureus, for which any growth” – Based on this and having in mind the EBJIS criteria, in your study a positive culture = 2CFU/ml – for a confirmed infection >50CFU/ml are needed and for likely > 1 CFU/ml. Also the authors of the EBJIS criteria report “Any positive culture from sonication fluid must be considered as a potential infection, but > 50 colony-forming units/ml (CFU/ml) confirms infection. The proposed cut-offs refer to a non-concentrated technique. If the concentration technique is applied, the suggested cut-off is 200 CFU/ml to confirm an infection. If other variations to the protocol are used, validated cut-offs for each protocol must be applied.”

- “removal of orthopedic implants for any reason.” Also “microbes” - Please rephrase in an more academic manner

- “By the 2021 EBJIS criteria, the most common organisms were Staphylococcus (51.8%) and gram-negative organisms constituted 15.6% of all pathogens.” As a personal opinion, the sentence should be rephrased to avoid confusion. Some might think that the EBJIS criteria are also criteria for the pathogens.

- Could you please confirm if there have been multiple revisions related to prosthetic joint infection caused by fungi or mycobacteria?

- “received antibiotics before operation” – as a treatment or perioperative prophylaxis?

6. PLOS authors have the option to publish the peer review history of their article (what does this mean?). If published, this will include your full peer review and any attached files.

Reviewer #1: No

Reviewer #2: No

---

## [Author Response · Author response to Decision Letter 0]

27 May 2024

Title: The microbiology of periprosthetic joint infections as revealed by sonicate cultures in Korea: Routine use of fungal and mycobacterial cultures is necessary?

Dear Editor-in-Chief

I would like to thank you for giving us the opportunity to revise our work. We have revised the manuscript in light of the valuable suggestions, and we believe it has been greatly improved. Attached you will find our revised manuscript, as well as our responses to the comments. This manuscript has not been published in any journal and is not under consideration for publication elsewhere. All authors including myself have seen and approved this manuscript.

Again, we thank you and the reviewers for a constructive and helpful review. 

Here are the detailed descriptions

Sincerely yours,

Kyung-Hwa Park

Department of Infectious diseases, 

Chonnam National University Medical School

42 Jebong-ro, Dong-gu, Gwangju, 61469, Korea

 

Responses to comments: 

Comments:

Reviewer #1: This is an important topic especially since we need a diverse voice in Ortho research. However, it needs further work before being ready for publication.

Major comments:

1. It’s confusing what are the primary objectives of this study. It seems like it is focusing on the performance and results of sonicate cultures, fungal and mycobacterial cultures. However, it also talks about the overall microbiology of PJI in Table 2, then sensitivity/specificity of microbiologic testing in Table 4. I recommend focusing on the paper on sonicate culture, fungal and mycobacterial culture results. For the gold standard, the authors could focus on EBJIS criteria instead of adding the ICM criteria. The accuracy of synovial and periprosthetic tissue cultures is another broad topic which can be addressed using another strategy and not appropriate to be linked to this paper.

→ We agree with reviewer’s comment absolutely. As recommended, we focused on results of sonicate cultures, fungal and mycobacterial cultures. Related paragraphs regarding the sensitivity/specificity of microbiological tests and Table 4 were deleted in method and results section of revised manuscript. Also, we focused EJBIS criteria and removed ICM criteria in the revised manuscript.

Minor comments:

1. Recommend editing Introduction and abstract to be more general in nature for broader readership rather than focusing on Korea. Suggest, “The use of sonicate cultures in clinical microbiology laboratories of tertiary general hospitals is not yet widespread. Orthopedic surgeons often request multiple sets of fungal, mycobacterial, and traditional bacterial cultures. Since the microbial yields from sonicate and tissue samples vary significantly across different medical centers and countries, further research is necessary to investigate the microbiology of PJI and the effectiveness of sonicate cultures”

→ As recommended, we have changed the description of the abstract and introduction section of revised manuscript. 

2. Edit terms “staphylococci” – Small letter s, not capital. “Gram-negative”- Capital letter G.

→ As recommended, capitalization errors in the revised manuscript were changed.

3. Harmonize "periprosthetic joint infections" instead of "prosthetic"

→ As you recommended, we harmonized the terms to “periprosthetic joint infection.

4. Sonication was not part of the IDSA diagnostic criteria, please revise.

→ As recommended, we have changed the description of the introduction section of revised manuscript.

5. Recommend changing term, “efficacy” to “accuracy” and “utility” in introduction. Diagnostic tests are not judged based on efficacy. It’s a term for therapeutic tests.

→ As recommended, we have changed the terms of the introduction section of revised manuscript.

Reviewer #2:

The manuscript titled "The microbiology of prosthetic joint infections as revealed by sonicate cultures in Korea: Routine use of fungal and mycobacterial cultures is necessary?" focuses on interesting topic. I commend the authors for their dedication and express my appreciation for the chance to evaluate their manuscript. The manuscript is crafted in a decent manner,that can in terms on English be also improved. Congratulations to the authors on their findings. A retrospective multi-centre study that in my opining should be taken into consideration to be published after a revision. The conclusions are consistent with the evidence and arguments presented and effectively address the main question posed.

After assessing the manuscript, the following issues raised my concerns or represent suggestions that from my point of view can increases the overall quality of the manuscript:

1. Abstract – results – “According to EBJIS criteria, total of 181 patients were diagnosed with acute (28 patients) or chronic (153 patients) PJIs” – The EBJIS criteria represent a set of definition criteria for periprosthetic joint infections (PJIs), whereas in 2004 for e.g., Zimmer et al proposed a classification system for PJIs, which was subsequently published in the New England Journal of Medicine. It is important to note that while the EBJIS criteria serve to define PJIs, the classification system proposed by Zimmer et al categorizes PJIs into distinct groups based on their clinical presentation and various other factors. Please also note the statements of the authors of the EBJIS criteria “This definition does not distinguish PJI on the basis of the duration of the infection (acute or chronic) or the time of onset from implantation (early or late). These terms are not defined with any degree of certainty with time-dependent cut-offs, and so cannot be included in a definition of PJI.”

→ We agree with reviewer’s comment absolutely. As recommended, we have changed the abstract of revised manuscript.

“According to EBJIS criteria, a total of 181 patients were diagnosed with PJI, and 141 patients received microbiological confirmation through sonicate fluid culture or tissue culture. Of the 181 patients, 28 were classified with acute PJI (within 3 months of implantation) and 153 with chronic PJI.”

2. Abstract – “In total, 11 cases cultured positive only in tissue culture, whereas 20 cases cultured positive only in sonicate culture.” What is your approach in cases of only positive sonication fluid cultures? As a suggestion – take a look at this multicenter retrospective cohort study - Rondaan C, Maso A, Birlutiu RM, Fernandez Sampedro M, Soriano A, Diaz de Brito V, Gómez Junyent J, Del Toro MD, Hofstaetter JG, Salles MJ, Esteban J, Wouthuyzen-Bakker M; ESCMID Study Group on Implant Associated Infections (ESGIAI). Is an isolated positive sonication fluid culture in revision arthroplasties clinically relevant? Clin Microbiol Infect. 2023 Jul 28:S1198-743X(23)00345-2. doi: 10.1016/j.cmi.2023.07.018. Epub ahead of print. PMID: 37516385.

→ Thank you for your kind comment. According to EBJIS criteria, we included an isolated positive sonicate culture pathogen. Among 20 cases cultured positive only in sonicate culture (not tissue), there were 16 cases with an isolated sonicate culture. We added this data in the results section of revised manuscript and Table 3. 

 “Among 20 cases, 16 cases were patients with an isolated positive sonicate culture”. 

We discussed this as limitation in the discussion section of the revised manuscript.. 

→ Lastly, there were 16 cases with an isolated positive sonicate culture and a significant number were low virulent microorganisms in our work. Although Rondaan C, et al. recently reported the clinical relevance of an isolated positive sonicate culture [32], these findings require further exploration for clinical importance. 

3. Diagnosis of PJIs- “To evaluate the diagnostic utility of periprosthetic tissue and sonicate cultures, PJIs were diagnosed using modifications of clinical criteria (excluding microbiological results) used in previous studies [12, 16].” Do you have some internal validation study for this criteria?

→ This registry is part of an ongoing cohort study for sonicate culture that began in 2016. The medical records were abstracted by two of the authors (orthopedic surgeon and infectious disease specialist). Although we did not validate this criteria internally, our data were reviewed by each specialists. 

We added these sentences in study design in the method section of revised manuscript. 

“We made the registry of sonicate culture since October 2016. Patients undergoing prostheses removal surgery, whose prostheses were sent for sonication between January 2017 and December 2022, were analyzed. The medical records were abstracted by two of the authors (orthopedic surgeon and infectious disease specialist).”

4. Sonication cultures – “For sonicate fluids, we considered that a culture was positive if growth exceeded 20 CFU/10 mL, with the exception of virulent microorganisms such as S. aureus, for which any growth” – Based on this and having in mind the EBJIS criteria, in your study a positive culture = 2CFU/ml – for a confirmed infection >50CFU/ml are needed and for likely > 1 CFU/ml. Also the authors of the EBJIS criteria report “Any positive culture from sonication fluid must be considered as a potential infection, but > 50 colony-forming units/ml (CFU/ml) confirms infection. The proposed cut-offs refer to a non-concentrated technique. If the concentration technique is applied, the suggested cut-off is 200 CFU/ml to confirm an infection. If other variations to the protocol are used, validated cut-offs for each protocol must be applied.”

→ Thank you for kind comment. We would like to quote the following paper published recently- Alvarez Otero J, Karau MJ, Greenwood-Quaintance KE, Abdel MP, Mandrekar J, Patel R. Evaluation of Sonicate Fluid Culture Cutoff Points for Periprosthetic Joint Infection Diagnosis. In Open Forum Infectious Diseases 2024 Mar 20 (p. ofae159). This study discussed ideal sonicate fluid culture cutoff points for PJI diagnosis and concluded that 20 CFU/10ml was appropriate for the diagnosis of hip and knee PJI according to Mayo Clinic protocol. 

→ Because we used Mayo Clinic protocol and considered 20 CFU/10ml as validated cut-offs, we added this as reference in the method of revised manuscript and mentioned this in S1 Table.

5. “removal of orthopedic implants for any reason.” Also “microbes” - Please rephrase in a more academic manner

→ As recommended, we have changed the description of the methods section of revised manuscript. We have changed microbes into microorganisms. 

“During the study period, 333 patients underwent orthopedic implant removal for various reasons including suspected PJI, aseptic loosening, or fixation failure.” 

6. “By the 2021 EBJIS criteria, the most common organisms were Staphylococcus (51.8%) and gram-negative organisms constituted 15.6% of all pathogens.” As a personal opinion, the sentence should be rephrased to avoid confusion. Some might think that the EBJIS criteria are also criteria for the pathogens.

→ As recommended, we have rephrased the sentence.

“Among microbiologically confirmed 141 PJI cases, the most common organisms were staphylococci (51.8%) and Gram-negative organisms constituted 15.6% of all pathogens.”

7. Could you please confirm if there have been multiple revisions related to prosthetic joint infection caused by fungi or mycobacteria?

→ We added this sentence in the results section of revised manuscript. 

“In 3 out of 5 cases in which mycobacteria were identified and in 7 out of 13 cases in which fungus was identified, multiple revision operations were performed.”

8. “received antibiotics before operation” – as a treatment or perioperative prophylaxis?

→ This refers to cases where antibiotics are used for therapeutic purposes, not prophylaxis. We have used term more clearly. 

“In our work, about 46% of patients received antibiotics treatment before operation and 70% of patients yielded ≤ 4 tissue samples.”

---

## [Decision Letter · Decision Letter 1]

2 Jul 2024

PONE-D-24-06916R1The microbiology of periprosthetic joint infections as revealed by sonicate cultures in Korea: Routine use of fungal and mycobacterial cultures is necessary?PLOS ONE

Dear Dr. Park,

Thank you for submitting your manuscript to PLOS ONE. After careful consideration, we feel that it has merit but does not fully meet PLOS ONE’s publication criteria as it currently stands. Therefore, we invite you to submit a revised version of the manuscript that addresses the points raised during the review process.

We look forward to receiving your revised manuscript.

Kind regards,

Abbas Farmany

Academic Editor

PLOS ONE

Journal Requirements:

Additional Editor Comments:

- Add a statistical analysis section in the M&M.

- Separate the conclusion section.

Reviewers' comments:

Reviewer's Responses to Questions

**Comments to the Author**

1. If the authors have adequately addressed your comments raised in a previous round of review and you feel that this manuscript is now acceptable for publication, you may indicate that here to bypass the “Comments to the Author” section, enter your conflict of interest statement in the “Confidential to Editor” section, and submit your "Accept" recommendation.

Reviewer #2: All comments have been addressed

2. Is the manuscript technically sound, and do the data support the conclusions?

Reviewer #2: Yes

3. Has the statistical analysis been performed appropriately and rigorously? 

Reviewer #2: Yes

4. Have the authors made all data underlying the findings in their manuscript fully available?

Reviewer #2: Yes

5. Is the manuscript presented in an intelligible fashion and written in standard English?

Reviewer #2: Yes

6. Review Comments to the Author

Reviewer #2: Dear Authors,

The revisions you have made in response to the comments from both myself and the other reviewer have significantly enhanced the quality of the manuscript. The improvements are evident and have addressed the key concerns raised during the review process.

From my perspective, the manuscript has now reached a standard that warrants further consideration for acceptance. I believe it will make a valuable contribution to the field.

Thank you for your diligent efforts in refining your work.

7. PLOS authors have the option to publish the peer review history of their article (what does this mean?). If published, this will include your full peer review and any attached files.

Reviewer #2: No

---

## [Author Response · Author response to Decision Letter 1]

2 Aug 2024

Title: The microbiology of periprosthetic joint infections as revealed by sonicate cultures in Korea: Routine use of fungal and mycobacterial cultures is necessary?

Dear Editor-in-Chief

I would like to thank you for giving us the opportunity to revise our work. We have revised the manuscript in light of the valuable suggestions, and we believe it has been greatly improved. Attached you will find our revised manuscript, as well as our responses to the comments. This manuscript has not been published in any journal and is not under consideration for publication elsewhere. All authors including myself have seen and approved this manuscript.

Again, we thank you and the reviewers for a constructive and helpful review. 

Here are the detailed descriptions

Sincerely yours,

Kyung-Hwa Park

Department of Infectious diseases, 

Chonnam National University Medical School

42 Jebong-ro, Dong-gu, Gwangju, 61469, Korea

 

Responses to edditor’s additional comments: 

#1: This is an important topic especially since we need a diverse voice in Ortho research. However, it needs further work before being ready for publication.

 → As recommended, we have added a statistical analysis section in the M&M.

#2: Separate the conclusion section.

 → As recommended, we have separated the conclusion section.

---

## [Editor Report · Decision Letter 2]

6 Aug 2024

The microbiology of periprosthetic joint infections as revealed by sonicate cultures in Korea: Routine use of fungal and mycobacterial cultures is necessary?

PONE-D-24-06916R2

Dear Dr. Park,

We’re pleased to inform you that your manuscript has been judged scientifically suitable for publication and will be formally accepted for publication once it meets all outstanding technical requirements.

Kind regards,

Abbas Farmany

Academic Editor

PLOS ONE
---

## [Editor Report · Acceptance letter]

8 Aug 2024

PONE-D-24-06916R2 

PLOS ONE

Dear Dr. Park, 

I'm pleased to inform you that your manuscript has been deemed suitable for publication in PLOS ONE. Congratulations! Your manuscript is now being handed over to our production team.

Kind regards, 

on behalf of

Dr. Abbas Farmany 

Academic Editor

PLOS ONE